# Summation Laws in Control of Biochemical Systems

**Hans V. Westerhoff** [1,2,3,4]

1   Department of Molecular Cell Biology, Amsterdam Institute for Molecules, Medicines and Systems, Vrije Universiteit Amsterdam, De Boelelaan 1108, 1081 HZ Amsterdam, The Netherlands; hvwesterhoff@gmail.com

2   School of Biological Sciences, Faculty of Biology, Medicine and Health, The University of Manchester, Manchester M13 9PT, UK

3   Swammerdam Institute for Life Sciences, University of Amsterdam, Sciencepark 904, 1098 XH Amsterdam, The Netherlands

4   Stellenbosch Institute of Advanced Studies (STIAS), Wallenberg Research Centre at Stellenbosch University, Stellenbosch 7600, South Africa

**Abstract:** Dynamic variables in the non-equilibrium systems of life are determined by catalytic activities. These relate to the expression of the genome. The extent to which such a variable depends on the catalytic activity defined by a gene has become more and more important in view of the possibilities to modulate gene expression or intervene with enzyme function through the use of medicinal drugs. With all the complexity of cellular systems biology, there are still some very simple principles that guide the control of variables such as fluxes, concentrations, and half-times. Using time-unit invariance we here derive a multitude of laws governing the sums of the control coefficients that quantify the control of multiple variables by all the catalytic activities. We show that the sum of the control coefficients of any dynamic variable over all catalytic activities is determined by the control of the same property by time. When the variable is at a maximum, minimum or steady, this limits the sums to simple integers, such as 0, $-1$, 1, and $-2$, depending on the variable under consideration. Some of the implications for biological control are discussed as is the dependence of these results on the precise definition of control.

**Keywords:** control coefficients; metabolic control analysis; systems biology; genomics; pharmacokinetic principles; systems biology and PBPK; time-dependent control analysis; systems pharmacology; growth rate; yield and efficiency

## 1. Introduction

The chemical networks in living organisms are all organized in the same hierarchy: a genome contains genes for proteins. Many of these are enzymes, i.e., catalysts of chemical (and transport) reactions in metabolism. The expression of a gene may be affected by addressing the DNA region upstream, by mutation or transcription factor. The effect is an altered level of the corresponding enzyme. At time scales that are important for physiology, i.e., function, intracellular metabolism is at a quasi-steady state. Hereby the metabolite concentrations and the fluxes become functions of the enzyme concentrations and hence of the gene activities. This organization has implications for the mathematics of the behavior of living cells, as we shall elaborate on below.

The sequencing of whole genomes, with the subsequent mapping of the roles that the subset of metabolic genes play in human metabolism [1], has led to an increased attention on how biological function may be adjusted by modulating gene expression or enzyme activities. With his background in both genetics and theoretical biochemistry, Henrik Kacser, together with James Burns, realized how gene copy number could determine biological function quantitatively. They identified the molecular basis of dominance [2]. The

explanation is that the sum over all the enzymes in a network, of their control coefficients on a flux, must equal one [3,4], and that there are many enzymes in the usual networks, so that the average control must be small (1/n for n enzymes in a linear pathway [2]). The fact that the sum of the flux control coefficients must equal one at steady state inspired Kacser and others to ask whether that control would then be distributed homogeneously among the enzymes or reside in a single key factor. For most biochemical networks, the answer appears to be: distributed but not such that all enzymes have the same small control on that flux, e.g., [5–13].

The importance of these results cannot be underestimated. For many years the concept that each metabolic pathway should have a single 'rate-limiting' step, preferably at its beginning, dominated both biochemistry and molecular biology [14–18]. Based on this concept, researchers searched for such rate-limiting enzymes, key genes, and key regulators, without finding these unequivocally. The number of oncogenes for instance, is not equal to 1, which would have been in accordance with the concept, but >100, with dire consequences for diagnosis and therapy [19]. Only relatively recently, has biomedical research begun focusing on drug combinations for the therapy of multifactorial diseases such as cancer, type-2 diabetes, and heart failure [20–22]. For microbiology the realization that the specific growth rate of microorganisms may be limited by multiple factors at the same time is of great importance for the development of new antibiotics in the context of multidrug resistance; and likewise for cancer [23,24].

At first, the summation laws were limited to metabolic fluxes and concentrations at steady state, with 1 and 0 for their respective sums. Subsequently, the laws have been extended to include time dependent metabolite concentrations and fluxes [25], as well as the time dependencies themselves [26,27]. Proofs of these laws were based on implicit differentiation [4,28], or on the theory of homogeneous functions [29,30]. The properties for which the laws have been derived remain limited in number. They do not include the specific growth rate, growth yield, and thermodynamic efficiency that are important for microbiology, for instance. Nor do they address the control of the transformation status of a tumor cell population, or the control of the area under the curve of the pharmacokinetics of a drug [31]. Here, we develop a novel mathematical proof of the summation laws, now for a multitude of new variables reflecting the dynamics of biochemical and biological networks. We base this proof on the concept of time-unit-invariance.

## 2. Results

### 2.1. The System

We will consider biochemical networks away from equilibrium in which various substances at well-defined concentrations (or mole numbers; we shall consider the volume of the well-stirred compartment to be fixed) are connected by (mostly) enzyme-catalyzed reactions. The latter may be chemical conversions or transmembrane transport processes. Each reaction *i* between reactant molecules ('substrates') forms a number of products and occurs at a certain reaction rate $v_i$. The boundaries of the system of the networks are set by fixed concentrations, fixed fluxes, or combinations of these. The concentrations and reaction rates within the network are dependent variables that depend on and reflect the state of the system. The latter is determined by all the fixed enzyme concentrations and other parameters (we shall call independent variables 'parameters'), such as the rate constants of the enzymes, their $k_{cat}$, their Michaelis and product inhibition constants, as well as by the fixed external concentrations or fluxes. We assume that the time evolution of the state variables is stable in the sense of Lyapunoff [32,33]. If its system parameters are left unaltered, the system considered here will evolve into a stationary state, but the analysis in this paper is not limited to such stationary states: it also addresses the time dependence of concentrations and of other state variables, e.g., after a parameter has undergone an instantaneous step change. In particular, we shall discuss the extent to which, at any point in time, the state variables change magnitude when any of the enzyme activities has

undergone a permanent infinitesimal modulation at time zero. We shall prove a number of laws that constrain the magnitudes of the corresponding [34] control coefficients.

### 2.2. Control Coefficients

A control coefficient quantifies the extent to which a catalytic process determines a system variable. Originally control coefficients of metabolite concentrations and metabolic fluxes were defined only for systems at steady state [3,4,34–36]. Their definition has since been generalized to time dependent metabolite concentrations and fluxes [25,37] and to properties characterizing time dependencies, such as cycling times, oscillation frequencies, half times, and transit times [26,27]. Here, we write the definition of the control coefficient of any dependent state variable $x(t)$ in a biochemical network as:

$$C_{e_j}^{x(t)} \stackrel{\text{def}}{=} \left( \frac{\partial \ln(x(t))}{\partial \ln(e_j)} \right)_{de_k=0 \ for \ all \ k \neq j} \tag{1}$$

$e_j$ refers to the catalytic activity of enzyme j (or of any other process if it is not enzyme-catalyzed); sometimes written as $v_j$ or as $V_{max,j}$. This definition applies to any variable x, which may be a concentration, a reaction rate, an electric potential, a time change of a concentration, or the area under the curve (AUC) of an intracellular toxin concentration, etc. All these together will populate the vector $\vec{x}(t)$, which we here denote by $x(t)$, with $t$ referring to time. We will here consider a set of, positive, x's that can vary independently of one another; pre-existing dependencies, such as through moiety conservation, must be removed by transformations [28,30].

The vector $\vec{e}$ (which will be denoted by $e$ below) represents the *complete* set of catalytic activities, each with a specificity $j$ for a chemical or transport reaction, that can be formulated explicitly and do not duplicate others: the $e$'s represent *all* parameters (independent variables) with time dimensionality $-1$ (see below). For more complex systems, for instance, with metabolite channeling, reaction steps involving multiple *proteins* may need to be replaced by a vector of the corresponding rate constants, but we shall not deal with such complications here [38,39]. The definition of the control coefficient should be interpreted in the sense of an agent $p_j$ specifically affecting enzyme activity $e_j$. The concentration of that agent is altered instantaneously at time zero, and the effect on the variable $x(t) > 0$ is determined. Therefore, more precisely:

$$C_j^{x(t)} = \left( \frac{\frac{\partial \ln(x(t))}{\partial p_j}}{\frac{\partial \ln(e_j)}{\partial p_j}} \right)_{de_k=0 \ for \ all \ k \neq j}. \tag{2}$$

The time ($t > 0$) coefficient is defined as:

$$C_t^{x(t)} = \left( \frac{\partial \ln(x(t))}{\partial ln(t)} \right)_{de_k=0 \ for \ all \ k} \tag{3}$$

### 2.3. Setting the Time

The time coefficient may be rewritten (for $t \geq 0$) as:

$$C_t^{x(t)} = \frac{t}{x(t)} \cdot \left( \frac{\partial x(t)}{\partial t} \right)_{de_k=0 \ for \ all \ k}. \tag{4}$$

This shows that for any actual development over time of the property $x$, $\frac{\partial x(t)}{\partial t}$ is well defined. The time coefficient defined above is not yet uniquely determined, however. This is because $t$ depends on the choice made by the observer of when $t$ should be called zero. Choosing that time point at 50 rather than 2 min before the time point $t$ would change the time coefficient of x by a factor of 25 (through the factor $t$ in the above equation). A similar

uncertainty exists for the property $x$. In order to remove these uncertainties, we consider the type of system that we address more closely: deterministic Markovian systems. These are systems of which the development over time after any given time point is unique and fully determined by the magnitudes (initial values) of a number of state variables (called $y$) of that system at that time point, plus two types of parameters. For these state variables $y$:

$$dy = f(k, q, y) \cdot dt. \tag{5}$$

We shall call the magnitudes of time and state variables at the initial time point $t_0$ and $y_0$, respectively (they may be set to zero later). In a metabolic system (biochemical network) at a given temperature and pressure, the properties $y$ are the concentrations of all the molecules indicated by the vector $y$. One type of parameter is represented by the vector $k$, which contains all parameters with dimension 1/time, including the rate constants and the enzyme activities. The vector $q$ represents all other parameters which do not have a time dimension, such as equilibrium and Michaelis–Menten constants, standard chemical potentials, and reaction stoichiometries. We assume that there are no parameters (independent variables) with other time dimensionalities than $-1$ or $0$. All these parameter values are considered to be fixed over the time span considered. When enzyme activities do depend on time, e.g., because of gene expression changes, this is dealt with by Hierarchical Control Analysis [40], or can remain part of the present analysis by describing them within $y$. The above expression is a generalization of the time dependence of metabolite concentrations in metabolic networks, which is described by:

$$\frac{d\vec{y}}{dt} = \mathbf{N} \cdot \vec{v} = \mathbf{N} \cdot \overrightarrow{E_i \cdot \varphi_i(\vec{y}, \vec{k}, \vec{K})} \tag{6}$$

with $\varphi$ a vector of enzyme rate laws, $k$ a vector of rate constants and $K$ a vector of Michaelis–Menten and equilibrium constants, $E$ is a corresponding vector of enzyme concentrations, $\mathbf{N}$ the matrix of reaction stoichiometries [30]. $\mathbf{N}$ and $K$ are now subsumed in $q$ and the vectors will be further represented by the corresponding scalars. Again, we drop the vector notation. Integration of the generalized equation then leads to:

$$y(t) = y_0 + \int_{t_0}^{t} f(k, q, y(t)) \cdot dt. \tag{7}$$

The state the deterministic system is in is fully defined by $y$, $k$ and $q$ and, thereby, by $y_0$, $k$, $q$, and $t - t_0$. Consequently, all other state functions $x$ are also determined by $y_0$, $k$, $q$, and $t - t_0$.

Here, we will further focus on what we call 'ideal biochemical networks' [38]. Every catalytic activity (including transport activities) in these is provided for by a specific protein (enzyme) and all these proteins function independently of each other. This means that there is neither substrate channeling nor group transfer between proteins (when there are such processes, the treatment becomes more complex but remains essentially the same). In such networks, the set of rate constants $k$, can be replaced by the smaller set of enzyme activities $e$. The initial concentrations can be added to the $q$ parameters to constitute the parameter set $p$, all without time dimension. Consequently, the change since time zero of any state variable $x$ is a function of the enzyme activity vector $e$, the vector of parameters without time dimension $p$, and the time $t - t_0$:

$$x - x_0 = x(e, p, t - t_0). \tag{8}$$

The parameter (independent variable) $t$ (and $t_0$ although not their difference) is ambiguous as its value at any occurrence of interest depends on the moment in history at which $t$ is taken to equal zero. Multiplying $d\ln(t)$ by $\frac{t}{t - t_0}$ one obtains the fully defined

property $\frac{dt}{t-t_0}$, which will not change when a different moment in time is taken as zero for the time axis:

$$dln(\tau) \stackrel{\text{def}}{=} \frac{t}{t-t_0} \cdot dln(t) = dln(t-t_0) = \frac{dt}{t-t_0}. \tag{9}$$

Consequently, one should substitute time parameter $\tau \stackrel{\text{def}}{=} t - t_0$ for time $t$ in the definition of the time coefficient, or [25] take $t_0 = 0$. This defines $t_0$ as the time at which the integration that was mentioned above starts and $x_0$ as the corresponding 'initial' magnitude of $x$: $x_0 \stackrel{\text{def}}{=} x(t_0)$. In summary, for the control coefficients to be unambiguously defined one should either resort to the definitions:

$$C_j^{\chi(\tau)} = \left( \frac{\partial \ln(\chi(\tau))}{\partial \ln(e_j)} \right)_{de_k=0 \ for \ all \ k \neq j, d\tau=0} \tag{10}$$

$$C_\tau^{\chi(\tau)} = \left( \frac{\partial \ln(\chi(\tau))}{\partial \ln(\tau)} \right)_{de_k=0 \ for \ all \ k} \tag{11}$$

with:

$$\chi(\tau) \stackrel{\text{def}}{=} x - x_0 \tag{12}$$

and

$$\tau \stackrel{\text{def}}{=} t - t_0, \tag{13}$$

or to the simpler definitions:

$$C_j^{x(t)} = C_{e_j}^{x(t)-x_0} \cdot \frac{x(t) - x_0}{x(t)} \tag{14}$$

$$C_t^{x(t)} = C_\tau^{x(t)-x_0} \cdot \frac{x(t) - x_0}{x(t)} \cdot \frac{t}{t - t_0} \tag{15}$$

with the *proviso* that $t$ refers to the time elapsed since $x$ equaled $x_0$, and $x$ refers to the value of $x$ minus $x_0$. The simplest approach is then to set both $x_0$ and $t_0$ to zero as was conducted by [25].

### 2.4. Summation Laws: Derivation

The observed magnitude of a system variable that does not have the dimension of time should not depend on the unit that the observer uses to measure time. The observed magnitude of a variable that does have a time dimension must depend on the time unit to the extent that is precisely in accordance with that dimensionality. In order to illustrate this, we consider two observers of the same natural phenomena. One observer measures the time $\tau$ in hours and the other, referred to by $\tau'$, measures it in minutes: $\tau'$ is 60 times larger than $\tau$ numerically, although the two times actually refer to the same moment:

$$\tau' = \lambda \cdot \tau \tag{16}$$

with $\lambda = 60\text{min/h}$ in the example. System properties $x$ may partially have a time dimension. Reaction rates, for instance, become 60 times smaller numerically when expressed in moles per minute than in moles per hour, although physically they remain the same. More in general for $x'$ expressed in the time unit of $t'$ and $x$ expressed in time units of $t$:

$$x' = \lambda^{\rho_x} \cdot x \tag{17}$$

$\rho_x$ represents the time dimensionality of $x$. For concentrations, thermodynamic efficiency, growth yield (as flux ratio), electric potentials, cell transformation state, cell population, and chemical potentials $\rho = 0$. For reaction rates, transport rates, and fluxes $\rho = -1$. For the half times, area under the curve (AUC), and mean residence time (MRT) much used

in pharmacology [31] $\rho = +1$. For 'area under the moment curve' (AUMC) $\rho = +2$. Some parameters also have time dimensionality: enzyme activities ($e$) and rate constants ($k$) have time dimensionality $\rho = -1$. The other parameters lack time dimensionality and are represented by $p$.

The second observer will find:

$$x' = f\left(e', p', t' - t'_0\right). \tag{18}$$

This will compare as follows with the observations made by the first observer:

$$\lambda^{\rho_x} \cdot x(t, e_j, p_k) = x' = x(\lambda \cdot \tau, e_j / \lambda, p_k). \tag{19}$$

Since this should be true for any value of $\lambda > 0$, we can equate the logarithmic derivative with respect to $\ln(\lambda)$ of the left-hand side of Equation (19) to the same derivative of the right-hand side of that same equation. After rearranging the equation, we find:

$$\rho_x = \frac{\partial \ln(x)}{\partial \ln(\lambda \cdot t)} \cdot \frac{\partial \ln(\lambda \cdot t)}{\partial \ln(\lambda)} + \sum_j \frac{\partial \ln(x)}{\partial \ln\left(\frac{e_j}{\lambda}\right)} \cdot \frac{\partial \ln\left(\frac{e_j}{\lambda}\right)}{\partial \ln(\lambda)} + \sum_k \frac{\partial \ln(x)}{\partial \ln(p_k)} \cdot \frac{\partial \ln(p_k)}{\partial \ln(\lambda)} = \frac{\partial \ln(x)}{\partial \ln(\lambda \cdot \tau)} - \sum_j \frac{\partial \ln(x)}{\partial \ln\left(\frac{e_j}{\lambda}\right)}. \tag{20}$$

For $\lambda = 1$, and after multiplication by $\frac{x - x_0}{x}$ (if $x \neq 0$ unless $x_0 = 0$; see above) this becomes the generalized summation law for time dependent control coefficients $C_j^x$ (we write $x$ for $x(\tau)$):

$$\rho_x \cdot \frac{x - x_0}{x} + \sum_j C_j^x = C_\tau^x. \tag{21}$$

For concentrations $y$, this implies that at any time point [25]:

$$\sum_j C_j^y = C_\tau^y, \text{ the sum being taken over all catalytic activities catalytic } e_j. \tag{22}$$

The same should be true for the control of thermodynamic efficiency, the control of flux ratios and concentration ratios, the control of electric and chemical potentials, etcetera. The sum over all enzymes (catalytic activities) in the network of the concentration control coefficients of any substance should herewith be positive when its concentration is on the increase with time, negative when it is on the decrease. When it is at its maximum or just steady that sum should be zero:

$$\sum_j C_j^{[y]} = C_\tau^{[y]} =_{min,max, or \, steady \, state} 0. \tag{23}$$

The formulation for steady state (Equation (23)) is the classical summation law for concentration control coefficients [3,4]. The same law applies to control of transmembrane electric potentials, phosphorylation potentials, and DNA supercoiling. For any reaction rate $v$ at time point $\tau$:

$$\sum_j C_j^v = C_\tau^v + \frac{v}{v - v_0}. \tag{24}$$

Assuming (see above) that at $t = t_0 = 0$ the system was started from a state at zero flux so that $v_0 = 0$, this becomes:

$$\sum_j C_j^v = C_\tau^v + 1. \tag{25}$$

This implies that when the flux is at a maximum (or minimum), or the system is at steady state, increasing all activities in proportion will increase the flux in the same proportion, because then $C_\tau^v = 0$. Only when the flux is decreasing with time at a time coefficient of $-1$, may such a collective increase in process activities leave the flux unaffected. The steady state case (Equation (25)) at $C_\tau^v = 0$ is the classical flux-control summation law [3,4].

Let the "area under the curve up to time point t" (AUCt; [31]) be the time integral of the variable concentration of a substance in the compartment of interest:

$$AUCt \overset{\text{def}}{=} \int_0^t y \cdot dt. \tag{26}$$

As this has the dimension of time, the summation law predicts:

$$\sum_j C_j^{AUCt} = C_\tau^{AUCt} - 1. \tag{27}$$

When all xenobiotic has left the body, time no longer affects the AUCt so that $C_\tau^{AUCt} = 0$. AUCt then becomes the AUC known in pharmacology, for which the sum of the control coefficients should equal $-1$:

$$\sum_j C_j^{AUC} = -1. \tag{28}$$

The "area under the first moment curve up to time point t" is defined by [31]:

$$AUMCt \overset{\text{def}}{=} \int_0^t y \cdot t \cdot dt \tag{29}$$

and has a time dimensionality of +2. Accordingly, its summation law reads:

$$\sum_j C_j^{AUMCt} = C_\tau^{AUCt} - 2. \tag{30}$$

As the xenobiotic leaves the system, the AUMCt also becomes constant in time and the summation law reads:

$$\sum_j C_j^{AUMC} = -2. \tag{31}$$

The mean residence time up to time point $t$ is defined by [31]:

$$MRTt \overset{\text{def}}{=} \frac{\int_0^t y \cdot t \cdot dt}{\int_0^t y \cdot dt} \tag{32}$$

and, thereby, has a time dimensionality of 1. Consequently:

$$\sum_j C_j^{MRTt} = C_\tau^{MRTt} - 1. \tag{33}$$

When all xenobiotic has left the system, the MRTt becomes the *mean* residence time MRT, and the total control exercised by the enzymes equals $-1$.

Considering the concentration-versus-time curve of a xenobiotic after its injection into the body, one may wonder at what time a certain concentration is reached. One then sees the time as a function of that concentration (and of all the enzyme activities). We consider a modulation of all enzyme activities by the same factor, so that for all $i$'s $dlne_i = dlne_1$. We allow for a simultaneous modulation of time to such an extent that there is no change in y. This leads us to:

$$0 = \left( \frac{\partial \ln(y)}{\partial \ln(\tau)} \right)_{de=0} \cdot \frac{(dln(\tau_y))_{dy=0}}{(dln(e_1))_{dy=0}} + \sum_j \left( \frac{\partial \ln(y)}{\partial \ln(e_i)} \right)_{dy=0} \cdot \frac{(dln(e_i))_{dy=0}}{(dln(e_1))_{dy=0}}$$
$$= \left( \frac{\partial \ln(y)}{\partial \ln(\tau)} \right)_{de=0} \cdot \left( \sum_j C_j^{\tau_y} + 1 \right) \tag{34}$$

where we have used the corresponding summation law as well as the expression:

$$(dln(\tau))_{dy=0} = \sum_j \left( \frac{\partial ln(\tau_y)}{\partial \ln(e_j)} \right)_{dy=0} \cdot (dln(e_j))_{dy=0} \tag{35}$$

and the definition of the control coefficients of the time at which the curve reaches $y$, i.e.,:

$$C_j^{\tau_y} \stackrel{\text{def}}{=\!=} \left( \frac{\partial ln(\tau_y)}{\partial \ln(e_j)} \right)_{dy=0}. \tag{36}$$

The general solution is that the sum of the control coefficients of the specific time point $(\tau_y)$ at which the curve reaches the concentration y equals $-1$:

$$\sum_j C_j^{\tau_y} = -1. \tag{37}$$

This will be true for any concentration of the substance, and, therefore, also for where it is half the maximum value, either on the increasing or on the decreasing slope, and where it is at the maximum.

## 3. Discussion

We have here derived a number of summation laws constraining the control of properties of metabolic and other networks in Biology. For control with respect to concentrations, these summation laws had been derived before by using different methodologies. [25] used the phenomenon that an acceleration of all processes by a factor should make everything happen in the same way but at a time point earlier by that factor. Our approach to changing the time unit in which the system is observed is similar to this. For the steady state summation laws, others and ourselves have used the property that steady state fluxes and concentrations are homogeneous functions of enzyme activities [29,30], performed the corresponding thought experiments [3], or have taken derivatives of the balance equation at steady state [4,28].

In the present paper, we have found summation laws for many more properties than the previous works had found, such as for read-outs of pharmacokinetics (AUMC), results of microbial growth experiments (yields), non-equilibrium thermodynamic properties (efficiencies), chemical potentials and Gibbs energy differences of reactions, and half times of dynamic changes in concentrations. In fact, the summation laws we proved here are valid for state variables of any time dimensionality $\rho$.

The summation laws have multiple implications for the control of dynamic phenomena [41]. We here mention only a few examples: When a concentration is at its time maximum, the corresponding summation law implies that this maximum concentration cannot just be determined by a single reaction activity in the system; there must be at least one additional controlling activity with an opposite sign. This is important for oncology, as it proves that the control over the phosphorylation state of an important 'onco-protein' must be shared by at least two other gene products. The steady state thermodynamic efficiency of microbial growth cannot be determined by a single process activity either. This is important because energy processes and microbial growth may be optimal in part in terms of growth rate, in part in terms of growth yield, and in part in terms of the thermodynamic efficiency of growth [30,42]. For the half times, the summation law implies that if one activity controls that time, then there must at least be one other activity in control unless a factor increase in the former causes a reduction in the half time by the same factor. This is important for the understanding of the control of dynamic signaling by the MAP kinase cascade [43]. The sums for the area under the curve $(-1)$ and for the 'area under the first moment curve up to time point t' $(-2)$ are again new findings and of great potential interest to pharmacology, where these variables are used to characterize the pharmacokinetics of clinical drugs [31]. When some enzyme activity is limiting for a biotechnological or medical process, one often tries to activate it. This will then reduce its control coefficient. The summation law has the implication that this automatically makes some other process more limiting, suggesting a second candidate for optimization [44].

Of course, the existence of the summation laws depends on the way 'control' is defined. It hinges on taking the double logarithmic derivative, which corresponds to the percentage

increase of the controlled variable resulting from a 1% activation of a process [30,34]. It also depends on limiting the controlling factors to the set of catalytic activities: the control coefficients are but a subset of all possible sensitivity coefficients. Examining systems in terms of more sensitivity coefficients than the control coefficients can lead to more insight into the why's and how's of their design and functioning [45–48], but not to these summation laws. Formulation of the control in terms of straight rather than log–log derivatives [28] changes the summation laws to equations that are so complex that their meaning may elude the biologically interested reader.

For the summations to lead to the fixed numbers found here, i.e., for the properties to become laws, the set of control coefficients (over which the sum is taken) should be complete. This completeness means that all parameters with time dimension should be subsumed in the set, either directly or indirectly, because their effects can be represented by a modulation of the enzyme activities [49]. The completeness is served by the advent of both genomics and systems biology with their interest in producing the complete genome and proteome of organisms [50] and with the vision of making *genome-wide* metabolic maps of a variety of organisms [1,51,52].

It is occasionally suggested that summation laws pertain to sums over all enzymes in the pathway of the flux or concentration under consideration. The above derivation shows that this is not so: the sum is over all the reaction activities in the entire network. Indeed, steps with major control may reside outside the pathway proper [53]. That the control may be distributed over the entire genome-wide network explains why so many genes exert so little control in biology, notwithstanding the ubiquitous myth that every biological function is determined by a single 'key' gene or enzyme. Where Kacser and Burns noted this for the control of fluxes [2,54], we may now generalize to all functional properties of complex Life, if not complex society [55].

Our results are important for Biology at large as they prove that general principles, such as invariance to unit transformation [56], are not confined to Physics. Many of these principles extend to Life sciences, with consequences that are illuminating and important for Biology and with instantiations that are less important for Physics and inorganic Chemistry. An example is the issue of whether the first irreversible step in a pathway is the rate-limiting step, with all subsequent steps being irrelevant for the control of pathway flux. The summation law states that there should indeed be a total rate limitation (i.e., the sum of flux control coefficients) of one but that this may be distributed over the pathway steps. For the simpler pathways in Physics and Chemistry, the distribution is such that all control is in the first irreversible step. In Biochemistry, control tends to reside in demand rather than supply [57], probably as a result of evolutionary selection for the organism's fitness through responsiveness to changes in workload. The way evolution has achieved this is by developing product inhibition of the enzymes [30], a phenomenon that is absent from inorganic Chemistry and Physics since they lack enzymes. More specifically, the present paper is important because it generalizes these summation laws to many properties (dependent variables) of biological interest, including cell cycle time, specific growth rate, growth yield, growth efficiency, transformation state, elasticity (response) towards a metabolite or signaling molecule, survival probability, DNA structure, and epigenetic state: as proven above, the numbers to which the corresponding control coefficients sum are given by the time dimensionality of the variables.

**Funding:** This research received no external funding.

**Data Availability Statement:** All material that could be shared is in this paper.

**Acknowledgments:** This paper is dedicated to the late Reinhart Heinrich and the late Henrik Kacser who have done so much for the understanding of biology and life, with their Metabolic Control Analysis that turns 50 this year.

**Conflicts of Interest:** The author declares no conflict of interest.

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
