# Peer review of "Summation Laws in Control of Biochemical Systems"

_mathematics, doi:10.3390/math11112473_

Round 1
Reviewer 1 Report
It was quite relevant and interesting as the author propose some new procedure for summation laws.
The topic chosen by author is an interesting topic as there were some gaps in the development of summation laws processes.
The paper well written and the text clear and easy to read. The conclusions consistent with the evidence and arguments presented.
Overall, I found this manuscript a good piece of research work in summation laws and control dynamics. Recommending the paper for the publication.
Author Response
It was quite relevant and interesting as the author propose some new procedure for summation laws. Response: Thank you.
The topic chosen by author is an interesting topic as there were some gaps in the development of summation laws processes. Response: Thank you.
The paper well written and the text clear and easy to read. The conclusions consistent with the evidence and arguments presented. Response: Thank you.
Overall, I found this manuscript a good piece of research work in summation laws and control dynamics. Recommending the paper for the publication. Response: Thank you.
Reviewer 2 Report
Author's work is based on nice idea but this paper lacks valid proof of proposed summation laws in biological systems.
Obviously the author has great knowledge and experience in this area and I would suggest him to publish two papers. One review paper and another paper in which the laws of summation laws will be proven with results of data calculations/simulations.
Unfortunately, I can not recommend this paper to be published.
Author Response
Author's work is based on nice idea but this paper lacks valid proof of proposed summation laws in biological systems. Response: Thank you for pointing this out. Proofs have been clarified; please see the new version's tracked changes
Obviously the author has great knowledge and experience in this area and I would suggest him to publish two papers. One review paper and another paper in which the laws of summation laws will be proven with results of data calculations/simulations. Response. The review paper can only be done once this paper has been accepted, of course. The review paper will also contain some simulations for biochemical systems..
Unfortunately, I can not recommend this paper to be published. Response: The reviewer does not make clear what the reason is behind this recommendation. There is nothing I can do except clarifying the proofs (see above) and editing the manuscript line by line for improved clarity, which is what I have done.
Reviewer 3 Report
The paper is well written and of interest to the PKPD modelling community.
Minor English corrections required.
Author Response
The paper is well written and of interest to the PKPD modelling community. Response: thank you.
Minor English corrections required. Response: manuscript was reread and corrected for English.
Reviewer 4 Report
The manuscript has the ambition to present new summation laws for biological systems. Part of the results were known before but are obtained with a different approach. The applications are versatile and important all along. Despite this, the survey actually would enjoy a limited audience, as it requires both mathematical competencies and biological understanding. That said, it is written for experts in the field and would hardly arouse the interest of a newcomer to the field or just a curious researcher. Even the title is a bit questionable. The author emphasizes the importance of the definitions when discussing control coefficients. However, the notion of a biological system remains not that well-defined. Gene expression and enzyme stimulation are molecular-level events. So do we consider proteins, protein bundles, cells or higher-level systems as biological systems? It would be very helpful to enumerate the equations – something typical for a mathematical paper – so that one could refer to them by number and not in a descriptive manner. The manipulations are not always easy to follow; certain procedures should be better explained. Also, there is some sloppiness in the writing. Thus, removing the arrow symbol from a vector is called an abbreviation, which it definitely isn’t, but is a notation, not shorthand either. “Variable” and “property” are used interchangeably, which is understandable but still inconsistent. The impact of the observer on the measurements is somewhat oddly placed and could lead the inexperienced reader to incorrect conclusions. While it is true that a change in the initial moment changes the read-out after a certain time interval, it is also true that at that point in time the state of the system will be different, consequently the initial conditions, which will obviously lead to a different dynamic problem with another solution. But that doesn’t mean that simply by changing the starting point you can change the dynamics as such.
I believe that the results deserve a better chance and the manuscript would benefit greatly from appropriate editing, expansion, and further explanations, especially as it is prepared as a commemoration of the work of outstanding scholars.
Author Response
The manuscript has the ambition to present new summation laws for biological systems. Part of the results were known before but are obtained with a different approach. The applications are versatile and important all along. Despite this, the survey actually would enjoy a limited audience, as it requires both mathematical competencies and biological understanding. That said, it is written for experts in the field and would hardly arouse the interest of a newcomer to the field or just a curious researcher.
Response: Thanks for the assessment. I now added ample explanation of biological relevance to the Introduction and Discussion sections. PLease look at the tracked chnages in the new version.
Even the title is a bit questionable.
Response: Summation laws in biological systems altered to 'Summation laws in control of biochemical systems '
The author emphasizes the importance of the definitions when discussing control coefficients. However, the notion of a biological system remains not that well-defined. Gene expression and enzyme stimulation are molecular-level events. So do we consider proteins, protein bundles, cells or higher-level systems as biological systems?
Response: Biological system is now defined in the first part of the Introduction
It would be very helpful to enumerate the equations – something typical for a mathematical paper – so that one could refer to them by number and not in a descriptive manner.
Response: done
The manipulations are not always easy to follow; certain procedures should be better explained. Also, there is some sloppiness in the writing. Thus, removing the arrow symbol from a vector is called an abbreviation, which it definitely isn’t, but is a notation, not shorthand either.
Response: Issues removed and text around equations clarified.
“Variable” and “property” are used interchangeably, which is understandable but still inconsistent.
Response: Thanks for pointing this out. I now make consistent use of 'variable'
The impact of the observer on the measurements is somewhat oddly placed and could lead the inexperienced reader to incorrect conclusions. While it is true that a change in the initial moment changes the read-out after a certain time interval, it is also true that at that point in time the state of the system will be different, consequently the initial conditions, which will obviously lead to a different dynamic problem with another solution. But that doesn’t mean that simply by changing the starting point you can change the dynamics as such.
Response: I clarify that it is not the initial conditions that is transformed for the proof but that the time unit of observation is altered. The part about the initial point is simply there to deal with a detail, important detail, but detail.
I believe that the results deserve a better chance and the manuscript would benefit greatly from appropriate editing, expansion, and further explanations, especially as it is prepared as a commemoration of the work of outstanding scholars.
Response: Done , with thanks for all your suggestions.
Round 2
Reviewer 2 Report
No substantial corrections in the revised manuscript.